# Comparative Effectiveness of First-Line Selpercatinib versus Standard Therapies in Patients with RET-Activated Cancers: An Exploratory Interpatient Analysis of LIBRETTO-001

**DOI:** 10.3390/cancers16010140

**Published:** 2023-12-27

**Authors:** Filippo De Braud, Barbara Deschler-Baier, John C. Morris, Francis Worden, Yimei Han, Urpo Kiiskinen, Min-Hua Jen, Scott S. Barker, Sylwia Szymczak, Adrienne M. Gilligan

**Affiliations:** 1Department of Oncology and Hematology-Oncology, University of Milan, 20133 Milan, Italy; filippo.debraud@istitutotumori.mi.it; 2Department of Internal Medicine, University Hospital Würzburg, 97080 Würzburg, Germany; deschler_b@ukw.de; 3Department of Endocrinology, The Mayo Clinic, Rochester, MN 55905, USA; morris.john@mayo.edu; 4Department of Internal Medicine, University of Michigan, Ann Arbor, MI 48105, USA; fworden@med.umich.edu; 5Value Evidence Outcomes, Eli Lilly and Company, Indianapolis, IN 46285, USA; han_yimei@lilly.com (Y.H.); kiiskinen_urpo@lilly.com (U.K.); m.jen@lilly.com (M.-H.J.); barker_scott_s@lilly.com (S.S.B.); szymczak_sylwia@lilly.com (S.S.)

**Keywords:** LIBRETTO-001, post hoc, selpercatinib, standard therapy, retrospective, effectiveness outcomes

## Abstract

**Simple Summary:**

There are limited studies comparing selpercatinib to standard treatments in previously treated patients with certain solid cancers that have alterations in the *RET* receptor. The aim of this post hoc comparative effectiveness analysis study was to compare investigator-assessed outcomes among patients with *RET*-altered solid tumors who were treated with selpercatinib but naïve to standard therapies versus those previously treated with only standard therapies. This study showed that selpercatinib use as frontline therapy is associated with improved outcomes versus standard therapies in patients with *RET*-altered cancers in the lung, thyroid, and other non-lung, non-thyroid organs in the body. Selpercatinib use was associated with a significantly lower risk of treatment discontinuation, starting a new systemic treatment (or death), or disease progression compared to standard therapies.

**Abstract:**

Selpercatinib is indicated for locally advanced/metastatic *RET*-activated solid tumors after progression or following prior systemic therapies. Until the recently published data from LIBRETTO-431 and LIBRETTO-531, there were limited effectiveness data comparing selpercatinib with other first-line treatments in *RET*-activated non-small cell lung cancer (NSCLC), medullary thyroid cancer (MTC), and thyroid cancer (TC). This study analyzed patient data from LIBRETTO-001 and compared the outcomes (time to treatment discontinuation {TTD}, time to next treatment or death {TTNT-D}, time to progression {TTP}, and the objective response rate {ORR}) of first-line selpercatinib (selpercatinib arm) use with the outcomes of first-line standard therapies in patients who then received selpercatinib in later lines of treatment (comparator arm). Overall, the first-line selpercatinib arm had a longer TTD, TTNT-D, and TTP versus the first-line comparator arm. The hazard ratios (HRs) for TTD were 0.29 (NSCLC), 0.15 (MTC), 0.08 (TC); for TTNT-D, the HRs were 0.48 (NSCLC), 0.11 (MTC), 0.09 (TC); and for TTP, the HRs were 0.54 (NSCLC), 0.15 (MTC), and 0.12 (TC). The ORR was higher for first-line selpercatinib versus the first-line comparator (NSCLC: 85.3% vs. 39.7%; MTC: 82.6% vs. 15.2%; and TC: 81.8% vs. 31.8%). First-line selpercatinib use is associated with improved outcomes compared to first-line comparator therapies for patients with advanced/metastatic *RET*-activated cancers.

## 1. Introduction

Alterations in the Rearranged during Transfection (*RET*) receptor are frequently found in several types of cancers [1]. *RET* fusions are observed in about 1–2% of non-small cell lung cancers (NSCLCs) and 5–10% of papillary thyroid cancers (TCs), and *RET* mutations occur in approximately 70% of patients with medullary thyroid cancer (MTC) [2,3]. Certain point mutations in *RET* or in-frame fusions of *RET* with various partners such as CCDC6-RET, NCOA4-RET, and KIF5B-RET can lead to *RET* kinase activation, resulting in oncogenic tumors [4,5]. Until the approval of selpercatinib (a first-in-class, highly selective, and potent *RET* inhibitor with central nervous system penetration) in May 2020, the treatment options for patients with *RET*-activated cancers were limited to standard therapies such as multikinase inhibitors (MKIs) or platinum chemotherapy +/− immune checkpoint inhibitors (ICIs) [6]. However, since MKIs were not designed to specifically target *RET*, their use is associated with low clinical benefit and significant off-target side effects, such as nausea, diarrhea, rash, and hypertension, in some patients with *RET*-activated cancers [5]. On the other hand, selpercatinib has high RET selectivity and its efficacy was established from a phase I/II LIBRETTO-001 single-arm clinical trial [2,3,7,8]. 

Results from the LIBRETTO-001 trial in patients pre-treated with standard therapies showed an objective response rate (ORR) of 61.5% for *RET* fusion-positive NSCLC, 77.6% for *RET*-mutated MTC, and 85.4% for *RET* fusion-positive TC [9,10]. In patients naïve to other therapies, the ORR was 82.6% and 95.8% among those with *RET* fusion-positive NSCLC and TC, respectively, while in patients with *RET*-mutated MTC naïve to cabozantinib and/or vandetanib, it was 84.5% [8,9]. Additional efficacy and safety data associated with selpercatinib in the first-line (1L) treatment of advanced NSCLC (aNSCLC) and MTC have been reported from two ongoing Phase 3 clinical trials (LIBRETTO-431 and LIBRETTO-531) [11,12]. In the LIBRETTO-431 trial, interim analysis results showed that patients on selpercatinib had a significantly longer progression-free survival (PFS) of 24.8 months versus 11.2 months with platinum-based chemotherapy ± ICI (pembrolizumab); the hazard ratio (HR) for progression or death was 0.47 (95% confidence intervals (CIs): 0.31 and 0.70, *p* < 0.001). In the LIBRETTO-531 trial, the median PFS was not reached for selpercatinib-treated patients with *RET*-mutant MTC who were naïve to cabozantinib/vandetanib versus 16.8 months for those treated with cabozantinib/vandetanib; the HR (95% CI) for disease progression or death was 0.28 (0.16 and 0.48, *p* < 0.001). The safety data were consistent with previously reported data [8,11,12,13,14,15,16]. 

While single-arm trials (such as LIBRETTO-001) are common in cancers with rare oncogenic drivers due to the lower number of patients available for recruitment, they do not provide comparative data or allow for a direct comparison with other therapies (for example, standard therapies) [17]. Given the encouraging results from the LIBRETTO-001 trial [2,3,8,13], there is a continued need to assess the effectiveness of selpercatinib in the 1L setting compared to other standard therapies. However, until the recently published head-to-head data comparing selpercatinib to standard therapies and MKIs from the LIBRETTO-431 and LIBRETTO-531 trials [11,12], the availability of comparable effectiveness data from external historical controls either from other clinical trials or real-world studies was limited. Furthermore, the adjusted comparisons to external controls are based on a strong assumption that study designs are identical. Differences between studies in terms of either the outcome of interest or study setting may exist, which can affect the reliability of comparisons coming from two different studies [18]. The use of a control arm from within the same trial is less likely to be subject to bias than affecting comparisons of arms from different studies due to strict inclusion/exclusion criteria that are common to all patients in the trial. The LIBRETTO-001 trial enrolled patients with RET-activated cancers treated with selpercatinib in various lines of therapy [17]. This created an opportunity to use historical data, that is, data on prior treatments such as standard therapies from patients who then received selpercatinib in later lines of (≥ second-line {2L+}) of treatment and compare their 1L outcomes to patients who were naïve to standard therapies and received selpercatinib in 1L. 

This study aimed to compare the investigator-assessed (INV) outcomes among patients with *RET* fusion-positive aNSCLC patients, *RET*-mutated MTC patients, and *RET* fusion-positive TC patients enrolled in LIBRETTO-001 who received 1L selpercatinib to the outcomes of patients who received standard therapies in 1L. 

## 2. Materials and Methods

This retrospective, comparative effectiveness study used data from the LIBRETTO-001 clinical trial (13 January 2023, data cut-off). LIBRETTO-001 is an ongoing phase I/II single-arm, multicenter, open-label, multicohort clinical trial of patients with *RET*-driven cancers conducted at 89 study sites in 16 countries [2,6]. The comparator arm, one each for the aNSCLC, MTC, and TC patient cohorts, was established using patients who were treated with standard therapies in 1L and received selpercatinib in later lines of therapy (2L+). The selpercatinib arm for the aNSCLC and TC cohorts included patients who were naïve to any standard therapy and patients who were naïve to cabozantinib and/or vandetanib for the MTC cohort (Figure 1 and Figure 2). 

In the aNSCLC and TC cohorts, the index date for the selpercatinib arm was the start date of 1L selpercatinib. In the MTC cohort, the index date for the selpercatinib arm was the date of initiation of the earliest line of selpercatinib treatment, as patients may have received other systemic treatments in earlier lines of treatment. The index date for the comparator arms in the aNSCLC and TC cohorts was the date of initiating 1L standard therapies (e.g., platinum-based chemotherapy in the aNSCLC cohort and lenvatinib/sorafenib in the TC cohort). In the MTC cohort, the index date for the comparator arm was the date of initiating 1L MKI (e.g., cabozantinib/vandetanib).

The study population included patients with *RET* fusion-positive aNSCLC, *RET*-mutated MTC, or *RET* fusion-positive TC who had progressed on or were not tolerant to standard therapy. Patients who declined standard therapy, for whom no standard therapies existed, or who were not candidates for standard therapy in the investigator’s opinion, were also eligible to be included for analysis in LIBRETTO-001. Details on the procedures followed in the LIBRETTO-001 trial are described elsewhere [8,13]. Some of the study inclusion criteria were age ≥ 18 years on index date (or ≥12 years, where allowed by regulatory authorities), measurable or non-measurable solid tumor according to Response Evaluation Criteria in Solid Tumors (RECIST) version 1.1, Eastern Cooperative Oncology Group (ECOG) performance status of 0–2 with no sudden deterioration two weeks prior to the index date, a life expectancy of ≥3 months with adequate organ function, and a corrected QT interval of ≤470 msec. Patients were excluded from the study in case of an occurrence of a targetable mutation in EGFR or rearrangements in ALK/ROS1 (for aNSCLC cohort), prior treatment with another selective *RET* inhibitor, and major surgery within four weeks prior to the index date or radiation therapy within one week prior to the index date. For the MTC cohort, additional exclusion criteria were the presence of clinically significant tracheal or esophageal disease, or complete encasement of large blood vessels (for example, the aorta or pulmonary artery) [13].

### 2.1. Study Measures and Definitions

#### Demographic and Clinical Variables

Patient characteristics measured on the index date included age, sex, race, ECOG performance status, stage at initial diagnosis, smoking status, brain metastases, time from the initial diagnosis to the index date, *RET* fusion status (aNSCLC and TC cohorts), and *RET* alteration (MTC cohort). 

The primary outcomes were investigator-assessed (INV) time-to-event outcomes. These included time-to-treatment discontinuation (TTD), time-to-next treatment or death (TTNT-D), time-to-progression (TTP), and the objective response rate (ORR). 

TTD was defined as the time from the index date of 1L for the aNSCLC and TC cohorts or the given line for the MTC cohort to the date of treatment discontinuation. The follow-up date was variable depending on the time-to-event outcome. For patients who did not have a discontinuation, TTD was censored on the last date of the trial treatment.TTNT-D was defined as the time from the index date (1L for the aNSCLC and TC cohorts and given line for the MTC cohort) to the start date of the next line of therapy or death, whichever occurred first. For patients whose treatment was ongoing, the TTNT-D was ended on the cut-off date. If treatment was discontinued, the TTNT-D was the time until the start of the next treatment. Otherwise, the TTNT-D was ended at the last long-term follow-up visit or study end date.TTP was defined as the time from the index date (1L for the aNSCLC and TC cohorts and given line for the MTC cohort) to the date of the INV-assessed progression date (PD). Otherwise, the TTP was ended on the analysis date of the end of the study or the date of the last evaluable disease assessment, whichever occurred first.The ORR was defined as the proportion of patients with a documented complete response or partial response (CR/PR) based on an INV-reported assessment.

### 2.2. Statistical Analysis

Descriptive analyses were performed for patient characteristics and were reported as frequencies and percentages for categorical variables and means with standard deviations (SD) or minimum/maximum for continuous variables. For time-to-event outcomes, the number, median, and 95% confidence intervals (CIs) were reported. Kaplan–Meier analyses were used to descriptively characterize the TTD, TTNT-D, and TTP between the comparator and selpercatinib arms after matching approaches. Within the Kaplan–Meier analysis, the log-rank test was conducted to evaluate the differences between the comparator and selpercatinib arms. A Cox proportional hazards regression model was used to compare the selpercatinib and the comparator arms in time-to-event outcomes, and their HR and 95% CI were estimated. The association between the exposure to treatment and the response rate was estimated in terms of an odds ratio (OR).

Outcomes were compared between the treatment arms in each cohort using genetic matching algorithm [19] with a 1:1 matching without caliper to estimate the average treatment effect (ATE). This type of matching uses a generalized Mahalanobis distance metric as the distance measure and includes weights for each covariate. It matches the propensity score and individual covariates by using an automated search algorithm to yield better-matched samples. The genetic matching method offers the least imbalance and bias between samples and is not vulnerable to the misspecification of the propensity score that is seen with propensity score matching or inverse probability of treatment weighting [20]. For this study, the target population of interest was those treated with selpercatinib; therefore, an ATE was estimated by weighting the comparator arm to match the selpercatinib arm. Other matching/weighting methods based on ATE were also performed (propensity score matching, propensity score-based weighting using a generalized boosted model or a logistic regression model; these analyses are not reported in this paper). Covariates used in the matching included age at index date, sex, race, ECOG performance status, stage at initial diagnosis, smoking status, histology, time since initial diagnosis to start of first-line treatment, and presence of brain metastases.

### 2.3. Research Ethics

Patient consent was waived as this was a non-interventional study based on the secondary use of data from LIBRETTO-001 and institutional review board approval was not needed.

### 2.4. Subgroup Analyses

Analyses were replicated for the following subgroups:

#### 2.4.1. NSCLC Cohort

Patients naïve to standard therapy (1L selpercatinib arm) compared to patients with only one prior line of standard therapy (1L comparator arm).Patients naïve to standard therapy (1L selpercatinib arm) compared to patients that received pembrolizumab plus pemetrexed-platinum chemotherapy in 1L (comparator arm).

#### 2.4.2. MTC Cohort

Patients naïve to any standard therapy (1L selpercatinib arm) compared to patients with only one prior line of standard therapy (1L comparator arm).Patients naïve to any standard therapy (1L selpercatinib arm) compared to patients treated with cabozantinib and/or vandetanib in 1L (comparator arm).

#### 2.4.3. TC Cohort

Patients naïve to standard therapy (1L selpercatinib arm) compared to patients who received lenvatinib or sorafenib in 1L (comparator arm). No matching was applied due to the limited sample size.

## 3. Results

### 3.1. RET Fusion-Positive aNSCLC Cohort

Of the 315 patients with *RET* fusion-positive aNSCLC included in this study, 69 (21.9%) patients were in the 1L selpercatinib arm, and 246 (78.1%) patients were in the 1L comparator arm. Patients in the 1L selpercatinib arm were significantly older than the 1L comparator arm (mean {SD} years: 62.1 {13.0} vs. 56.9 {11.4} years, *p*-value = 0.0014; Table 1). Asians were the largest race group in the 1L comparator arm (47.8%) and second largest in the 1L selpercatinib arm (18.8%); black/African Americans comprised about 5% or less in each arm. Most patients (>94%) in both study arms had stage IIIB-IV NSCLC at initial diagnosis and ECOG performance status 0/1. Of the 246 patients in the comparator arm, 181 (73.3%) received chemotherapy in 1L. Chemotherapy in combination with anti-PD/PD-L1 was used by 40 patients (16.2%), while the remaining 10.5% received either anti-PD/PD-L1 monotherapy or MKI/tyrosine kinase inhibitor (TKI) (Appendix A). After matching, there were 68 patients in each arm; the baseline characteristics between the two arms were similar (Table 1).

The 1L selpercatinib arm had significantly longer median TTD, TTNT-D, and TTP and a higher ORR versus the 1L comparator arm. The median TTD in the selpercatinib arm was 25.3 months (95% CI: 17.4–32.6), significantly longer than the comparator arm (3.7 months {95% CI: 3.0–6.0}; HR: 0.29 {95% CI: 0.20, 0.43}; Figure 3). Throughout the study period, the probability of treatment discontinuation remained lower for patients in the 1L selpercatinib arm versus the 1L comparator arm.

The median TTNT-D in the 1L selpercatinib arm was 31.0 months (95% CI: 21.5–41.7) versus 14.6 months (95% CI: 10.5–16.7) in the 1L comparator arm (HR: 0.48 {95% CI: 0.32, 0.71}; Figure 4). The median TTP in the 1L selpercatinib arm was 22.0 months (95% CI: 13.9–28.4) versus 12.5 months (95% CI: 8.0–16.0) in the 1L comparator arm (HR: 0.54 {95% CI: 0.37, 0.78}; Figure 5). Patients in the 1L selpercatinib arm were significantly more likely to have a response than those in the 1L comparator arm (85.3% vs. 39.7%; OR: 8.8 {95% CI: 3.9, 20.2}).

#### Subgroup Analyses

Prior to matching in subgroup 1 (1L selpercatinib arm vs. those with one prior line of standard therapy {comparator arm}), the baseline characteristics were well-balanced between the two treatment arms, except for race where a higher proportion of patients in the selpercatinib arm were white (62.3% vs. 55.6%, *p*-value = 0.0098; Appendix A). Patients in the 1L selpercatinib arm had a significantly longer TTD and TTNT-D and were more likely to achieve a response versus the comparator arm. No differences were observed for the TTP across the two groups (Appendix A). In subgroup 2 (selpercatinib arm vs. those received pembrolizumab plus pemetrexed-platinum chemotherapy in 1L {comparator arm}), after matching, the baseline characteristics were well-balanced, except for sex where there was a higher proportion of female patients in the selpercatinib arm (63.2% vs. 45.6%, *p*-value = 0.0388; Appendix A). Findings were similar to the primary analysis where the selpercatinib arm had a significantly longer TTD, TTNT-D, and TTP, and the odds of the ORR were also higher in the selpercatinib arm versus the comparator arm (Appendix A).

### 3.2. RET-Mutated MTC Cohort

Of the 279 patients with *RET*-mutated MTC included in the study, 143 (51.3%) patients were in the 1L selpercatinib arm, and 136 (48.7%) patients were in the 1L comparator arm. Patients in the 1L selpercatinib arm were slightly older (mean {SD}: 55.7 {15.2}) years versus the 1L comparator arm (53.8 (15.8) years; a greater proportion in both arms were male (58.0%; 64.7%), although these differences were not statistically significant. Whites comprised the largest racial group in both groups (>86%) (Table 2). The median time from the initial diagnosis of MTC to 1L initiation was significantly longer in the 1L selpercatinib arm versus the 1L comparator arm (103.7 months vs. 81.2 months, *p* < 0.0002). Similar to what was observed in the aNSCLC cohort, most patients in both study arms of the MTC cohort had stage IV at initial diagnosis (1L selpercatinib arm: 48.2%, 1L comparator arm: 54.9%), and > 90% had an ECOG performance status 0/1. The 1L selpercatinib arm had significantly fewer patients with brain metastases compared to the 1L comparator arm (7.4% vs. 2.1%; *p*-value = 0.0374). In the comparator arm, 92 patients (67.7%) received vandetanib in 1L and the remaining 32.4% received cabozantinib (Appendix A). The baseline characteristics were similar in the two treatment arms (138 patients in each arm) after matching (Table 2).

Patients in the 1L selpercatinib arm had a significantly longer median TTD, TTNT-D, and TTP and higher ORR versus the 1L comparator arm. The median TTD was not reached in the 1L selpercatinib arm versus 16.6 months (95% CI: 12.6–18.6) in the 1L comparator arm (HR: 0.15 {95% CI: 0.10, 0.21}; Figure 6). The median TTNT-D was not reached for the 1L selpercatinib arm versus 18.9 months (95% CI: 16.1–22.4) in the 1L comparator arm (HR: 0.11 {95% CI: 0.07, 0.16}; Figure 7). The median TTP was not reached in the 1L selpercatinib arm versus 18.3 months (95% CI: 14.7–21.5) in the 1L comparator arm (HR: 0.15 {95% CI: 0.10, 0.22}; Figure 8). Patients in the 1L selpercatinib arm were significantly more likely to have a response than patients in the 1L comparator arm (82.6% vs. 15.2%, OR: 26.5 {95% CI: 14.0, 50.2}).

#### Subgroup Analyses

In subgroup 1 (1L selpercatinib arm vs. those with one prior line of standard therapy {comparator arm}), prior to matching, the baseline characteristics were well-balanced between the treatment arms, except for a higher proportion of patients with brain metastasis in the comparator arm (9.6% vs. 2.6%, *p* = 0.0363; Appendix A). Prior to matching in subgroup 2 (selpercatinib arm vs. those who received cabozantinib/vandetanib {comparator arm}), the baseline characteristics were well-balanced between the treatment arms (Appendix A). Results for the INV outcomes in subgroups 1 and 2 were similar to the primary analysis; patients in the 1L selpercatinib arm had a significantly longer TTD, TTNT-D, and TTP and were more likely to achieve a response compared to patients in the 1L comparator arm (Appendix A).

### 3.3. RET Fusion-Positive TC Cohort

A total of 65 patients with *RET*-fusion positive TC were included in this study, of which 24 (36.9%) were in the 1L selpercatinib arm and 41 (63.1%) were in the 1L comparator arm. Prior to matching, the baseline characteristics were well-balanced between the two treatment arms, except for race where a higher proportion of patients in the selpercatinib arm were white (75.0% vs. 58.5%, *p* = 0.0196). Additionally, the comparator arm had a significantly higher proportion of patients with brain metastases (4.2 vs. 29.3%, *p*-value = 0.0146; Table 3). The time from the initial diagnosis to the start of 1L was significantly longer for the selpercatinib arm versus the comparator arm (124.5 {154.8} months vs. 48.7 {109.2} months, *p*-value < 0.05). Among the 41 patients in the comparator arm, 22 patients (53.6%) received ‘other’ therapy in 1L. Of these, 21 patients (47.7%) received radioactive iodine (RAI) and one patient received everolimus. A total of 15 patients received either MKI or MKI plus RAI (36.6%), of which 13 patients received lenvatinib and 2 received sorafenib. The remaining four (9.8%) patients received either chemotherapy or chemotherapy plus other (Appendix A). The baseline characteristics were similar between the two treatment arms (22 patients in each arm) after matching (Table 3).

The 1L selpercatinib arm had a significantly longer median TTD, TTNT-D, and TTP and higher ORR versus the 1L comparator arm in patients with *RET* fusion-positive TC. The median TTD in the 1L comparator arm was 11.8 months (95% CI: 8.9–12.0) compared to not reached in the 1L selpercatinib arm (HR: 0.08 {95% CI: 0.03, 0.26}; Figure 9). The median TTNT-T was not reached in the 1L selpercatinib arm versus 10.4 months (95% CI: 9.0–29.2) in the 1L comparator arm (HR: 0.09 {95% CI: 0.03, 0.28}; Figure 10). The median TTP was not reached in the 1L selpercatinib arm versus 10.4 months (95% CI: 9.0–29.2 months; HR: 0.12 {95% CI: 0.04, 0.32}) in the 1L comparator arm (Figure 11). Patients in the 1L selpercatinib arm were significantly more likely to have a response than patients in the 1L comparator arm (81.8% vs. 31.8%; OR: 21.0 {95% CI: 1.5, 293.3}).

#### Subgroup Analyses

In subgroup 1 (1L selpercatinib arm vs. patients who received 1L lenvatinib/sorafenib {comparator arm}), without any adjustment, the baseline characteristics were well-balanced between the treatment arms, except for race which had a higher proportion of white patents in the selpercatinib arm (75.0% vs. 33.3%, *p* < 0.001; Appendix A). The unadjusted results were similar to the primary analysis where the selpercatinib arm had a significantly longer TTD, TTNT-D, and TTP, and patients were more likely to achieve a response compared to the comparator arm (Appendix A).

## 4. Discussion

In this post hoc, interpatient comparative effectiveness study, patient cohorts from the LIBRETTO-001 trial were used to compare investigator-assessed outcomes associated with 1L selpercatinib versus 1L standard therapies in the frontline treatment of *RET* fusion-positive aNSCLC, *RET*-mutated MTC, and *RET* fusion-positive TC. The results of these analyses showed that 1L selpercatinib use was associated with significantly improved outcomes (regarding TTD, TTNT-D, TTP, and ORR) compared to 1L standard therapies. More than half of the patients in the 1L selpercatinib arm continued treatment with selpercatinib at two years of follow-up in the aNSCLC cohort. In the MTC cohort, more than four out of five patients using 1L selpercatinib continued therapy after two years of follow-up, and nearly a third of selpercatinib-treated patients with MTC continued therapy at four years of follow-up. In the TC cohort, close to 60% continued therapy after two years of follow-up. Overall, the risk of treatment discontinuation was significantly lower among 1L selpercatinib-treated patients in all cohorts versus those on 1L standard therapies. Reasons to assess the cause of a lower risk of discontinuation with selpercatinib use were not in the scope of the present study. 

In this study, patients had a longer median TTNT-D on 1L selpercatinib treatment versus 1L standard therapies/MKIs. Greater than half of the patients (53%) in the aNSCLC cohort, about 88% in the MTC cohort, and nearly 60% in the TC cohort remained on 1L selpercatinib treatment without switching to another therapy after two years from initiation. A sizeable proportion of patients with aNSCLC, MTC, and TC did not have any disease progression after two years of continuous treatment with 1L selpercatinib, further demonstrating the clinical effectiveness of 1L selpercatinib versus 1L comparator therapies. The present study’s findings of selpercatinib’s clinical effectiveness in 1L settings versus standard therapies/MKIs are in line with a recent study that compared the outcomes of external controls from real-world data (real-world control arm) and clinical trial data (clinical trial control arm) of patients who were pre-treated with standard therapies to the outcomes of treatment-naïve patients from LIBRETTO-001 (selpercatinib arm) [17]. In this study, the authors reported that the selpercatinib arm had an improved ORR (87.2%) versus the real-world control in all entropy-balanced comparisons (range: 58.6–60.5%; *p* < 0.0001). Although the median overall survival (OS) was not reached for the selpercatinib arm in their study, patients with aNSCLC in the selpercatinib arm had a lower risk of death versus the real-world control (range of estimated HR = 0.25–0.38) [17]. Furthermore, while the median PFS was not reached for the selpercatinib arm versus the clinical trial control arm (pre-treated with platinum chemotherapy plus ICIs) in this study [17], the range of estimated HR for PFS was comparable (0.34–0.51) to the HR for the TTP (0.54) in our study. In addition, the estimated HR for PFS assessed using a genetic algorithm for 1L selpercatinib versus the clinical trial control arm (0.51, *p* = 0.045) approximated the HR for the TTP in our study (0.54). Evidence supports the use of PFS/TTP as a surrogate for OS [21]. 

Patients with aNSCLC have unmet therapeutic needs for effective therapies that should be used earlier on in treatment [22,23,24]. This was demonstrated in a real-world study where greater than 40% of patients with aNSCLC who received 1L chemotherapy or immunotherapy did not survive beyond 1L treatment [22]. This unmet need also exists for patients with *RET*-mutated MTC who received approved therapies with limited *RET* inhibitory activity such as cabozantinib or vandetanib [12]. Selpercatinib is a more effective treatment compared to platinum-based chemotherapy plus ICIs or docetaxel in patients with *RET* fusion-positive aNSCLC, or cabozantinib/vandetanib in patients with *RET*-mutated MTC, as demonstrated by the results from clinical studies [10,11,12]. Data from the LIBRETTO-431 trial have demonstrated that selpercatinib should be administered as the first-line treatment over standard therapies for patients with *RET* fusion-positive aNSCLC [11]. Similarly, the LIBRETTO-531 trial has shown that selpercatinib use was associated with improved PFS and treatment failure-free survival versus MKIs in patients with *RET*-mutated MTC [10]. In our study, the findings associated with the clinical outcomes related to the TTD, TTP, TTNT-D, and ORR favoring selpercatinib versus standard therapies or MKIs also support selpercatinib’s role as the frontline therapy for *RET*-activated NSCLC and MTC. Evidence from clinical studies and a recent meta-analysis have demonstrated selpercatinib’s safety in patients with *RET* fusion-positive aNSCLC [25,26,27]. In these studies, the occurrence of treatment-related adverse events as well as treatment discontinuation was lower with selpercatinib. In contrast, therapies such as MKIs had limited *RET*-specific inhibition and modest activity in *RET*-activated tumors. Furthermore, increasing the dosage of MKIs to improve activity may lead to substantial off-target adverse events and then to treatment discontinuation [5]. 

In another comparative effectiveness study that performed a matching adjusted indirect comparison of selpercatinib versus cabozantinib in a cohort of patients with *RET*-mutation positive MTC, the ORR for selpercatinib was 82.9% versus 31.7% for cabozantinib (OR {95% CI}: 10.5 {6.2, 17.6; *p* < 0.0001}) [28]. The conclusions from this study are similar to our findings (82.6% vs. 15.2%; OR {95% CI}: 26.5 {14.0, 50.2}) in that selpercatinib treatment was associated with significantly higher response rates. The interim results from the ongoing LIBRETTO-531 trial have also demonstrated improved outcomes in terms of the ORR favoring selpercatinib over cabozantinib and vandetanib (69.4% vs. 38.8%; OR {95% CI}: 3.7 {2.2, 6.3; *p* < 0.0001}) as well as better PFS (HR {95% CI}: 0.28 {0.16, 0.48; *p* < 0.001}) [10]. The interim efficacy results from the LIBRETTO-431 trial have also shown that selpercatinib use was associated with superior PFS vs. chemotherapy plus pembrolizumab (HR {95% CI}: 0.47 {0.31, 0.70; *p* < 0.0001}) as well as clinically meaningful improvements in the ORR [11].

### Strengths and Limitations

There is limited availability of external control cohorts from real-world or clinical trials that have reported treatment outcomes in *RET*-activated tumors. Additionally, regardless of the best analytical methods to adjust for differences in the observed population characteristics, differences between study designs may affect the validity of comparisons between two studies. Therefore, finding controls within the same trial with strictly defined inclusion criteria for a small target population could reduce the unobserved heterogeneity between the treatment and control cohorts. The present study used an interpatient analytical study design to create a control cohort of selpercatinib-naïve patients using historical 1L treatment outcome data from pre-treated patients within the LIBRETTO-001 study. This design fills a gap in the research for a comparative analysis between standard therapies and selpercatinib. Despite the strengths, we acknowledge some limitations. In the LIBRETTO-001 trial, tumor response was independently reviewed using RECIST criteria while in the real world, these criteria were investigator-assessed. Therefore, similar to what was noted as a limitation by Rolfo et al., 2022 [17], our findings related to TTP should be interpreted with caution. Since historical data from patients enrolled in LIBRETTO-001 were used in this study design, we could not assess PFS or OS as the data did not include information on events such as death. Furthermore, our estimates of selpercatinib’s effectiveness versus standard therapies are likely to be conservative given that patients in the control arm had completed at least one prior line of therapy and were healthy enough to be enrolled into the LIBRETTO-001 trial.

## 5. Conclusions

In this post hoc interpatient comparative effectiveness study, 1L selpercatinib demonstrated improved outcomes associated with clinical effectiveness compared to 1L standard therapies in patients with *RET* fusion-positive aNSCLC, *RET*-mutated MTC, and *RET* fusion-positive TC. Selpercatinib use was associated with a significantly lower risk of treatment discontinuation, starting a new systemic treatment (or death), or disease progression compared to standard therapies. Findings from this study suggest that 1L selpercatinib use may be more efficacious and more appropriate compared to standard therapies for patients with *RET*-activated cancers. Our findings were also supported by interim results from the ongoing LIBRETTO-431 and LIBRETTO-531 trials [11,12].

## Figures and Tables

**Figure 1 cancers-16-00140-f001:**
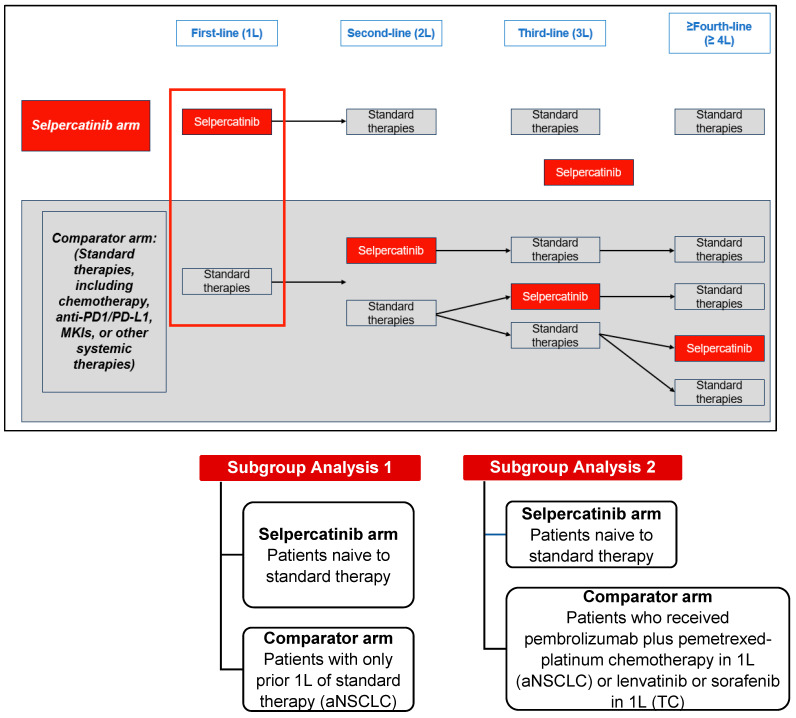
Study design of patients with RET fusion-positive aNSCLC or RET fusion-positive TC. Abbreviations: MKI, multikinase inhibitor; PD1/PD-L1, programmed death-1/programmed death ligand-1.

**Figure 2 cancers-16-00140-f002:**
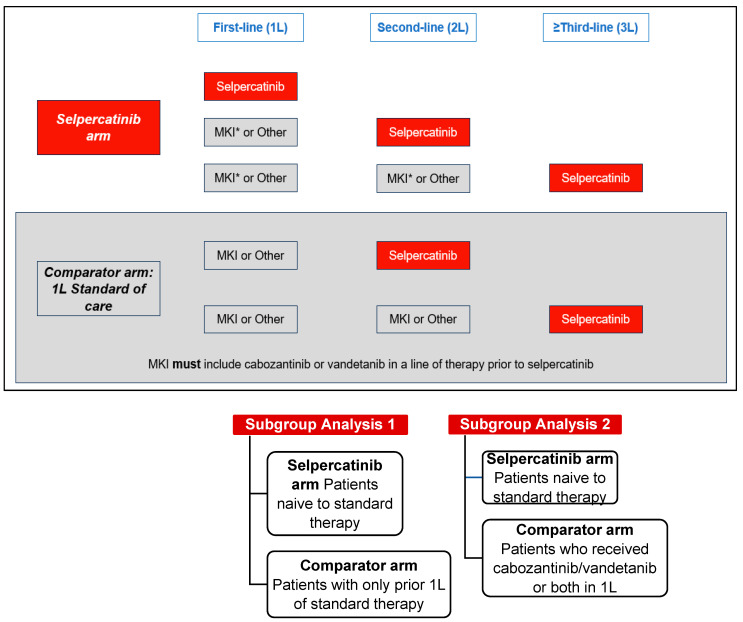
Study design of patients with RET-mutated MTC. Abbreviations: MKI, multikinase inhibitor. Note: MKI* in the selpercatinib arm included MKIs other than cabozantinib or vandetanib prior to selpercatinib.

**Figure 3 cancers-16-00140-f003:**
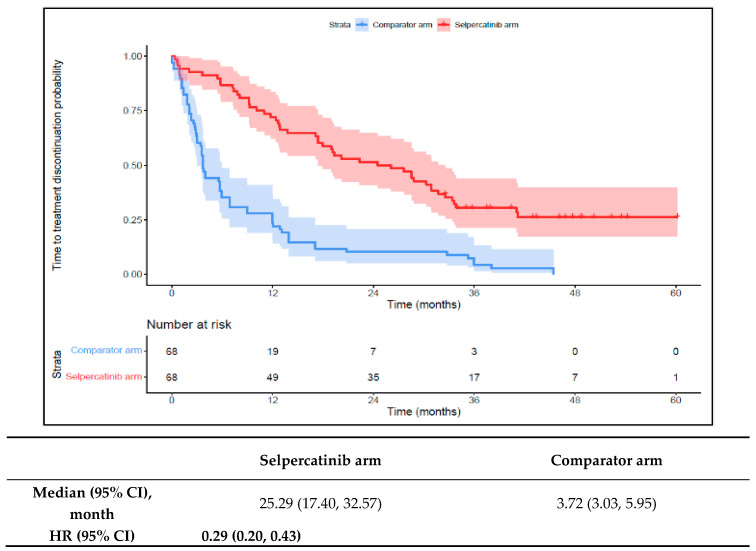
Kaplan–Meier estimates of the median TTD in the aNSCLC cohort. Abbreviations: aNSCLC, advanced non-small cell lung cancer; CI, confidence interval; and HR, hazard ratio. Note: The text in bold indicates statistical significance.

**Figure 4 cancers-16-00140-f004:**
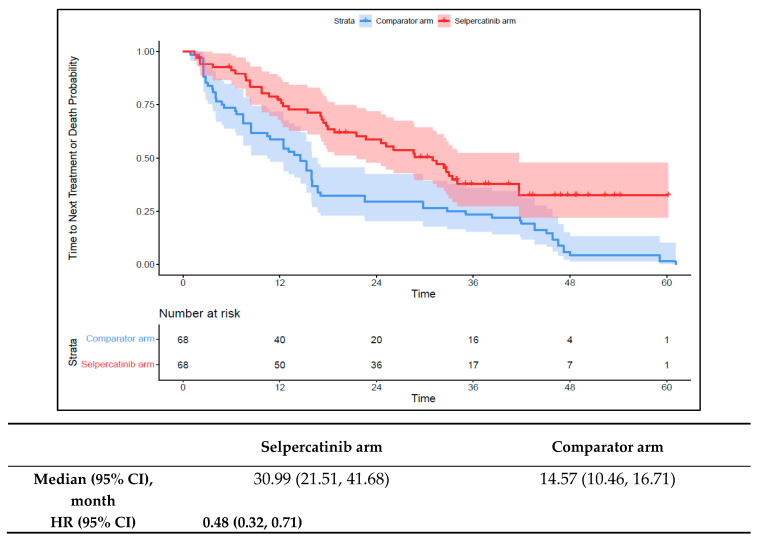
Kaplan–Meier estimates of the median TTNT-D in the aNSCLC cohort. Abbreviations: aNSCLC, advanced non-small cell lung cancer; CI, confidence interval; and HR, hazard ratio. Note: The text in bold indicates statistical significance.

**Figure 5 cancers-16-00140-f005:**
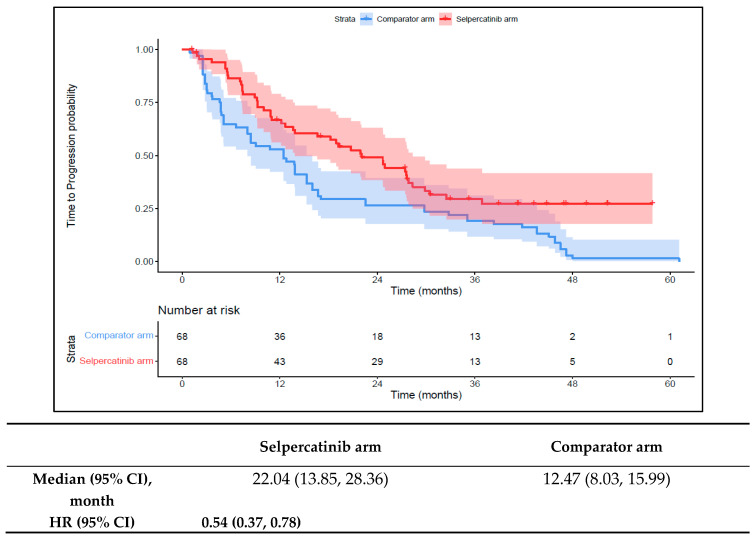
Kaplan–Meier estimates of the median TTP in the aNSCLC cohort. Abbreviations: aNSCLC, advanced non-small cell lung cancer; CI, confidence interval; and HR, hazard ratio. Note: The text in bold indicates statistical significance.

**Figure 6 cancers-16-00140-f006:**
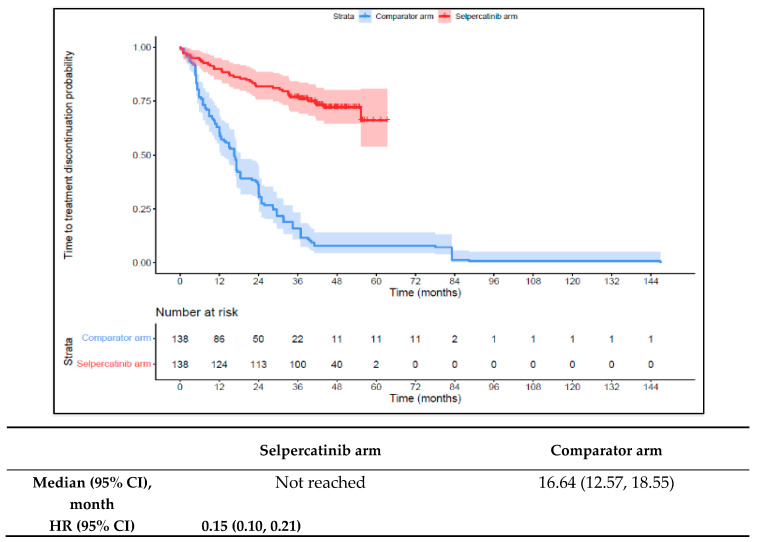
Kaplan–Meier estimates of the median TTD in the MTC cohort. Abbreviations: CI, confidence interval; HR, hazard ratio; and MTC, medullary thyroid cancer. Note: The text in bold indicates statistical significance.

**Figure 7 cancers-16-00140-f007:**
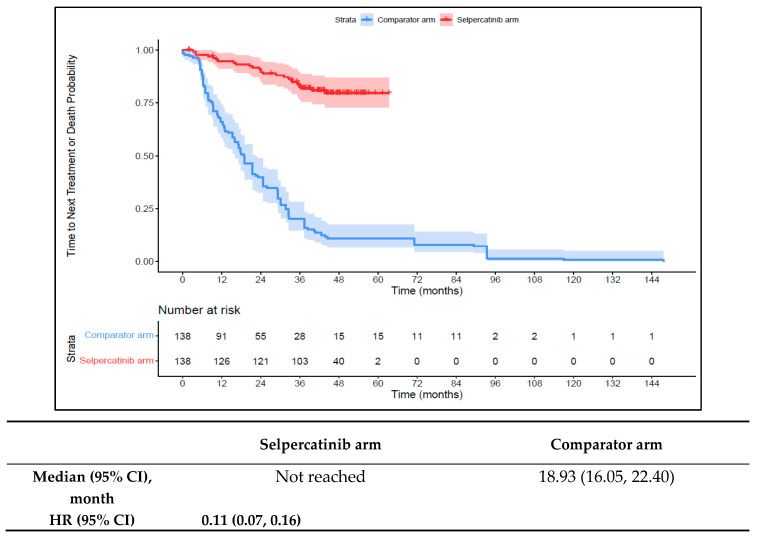
Kaplan–Meier estimates of the median TTNT-D in the MTC cohort. Abbreviations: CI, confidence interval; HR, hazard ratio; and MTC, medullary thyroid cancer. Note: The text in bold indicates statistical significance.

**Figure 8 cancers-16-00140-f008:**
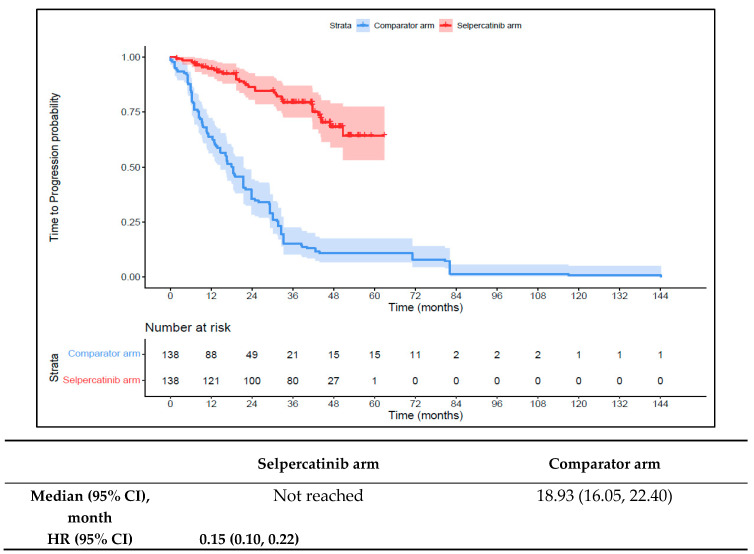
Kaplan–Meier estimates of median TTP in the MTC cohort. Abbreviations: CI, confidence interval; HR, hazard ratio; and MTC, medullary thyroid cancer. Note: The text in bold indicates statistical significance.

**Figure 9 cancers-16-00140-f009:**
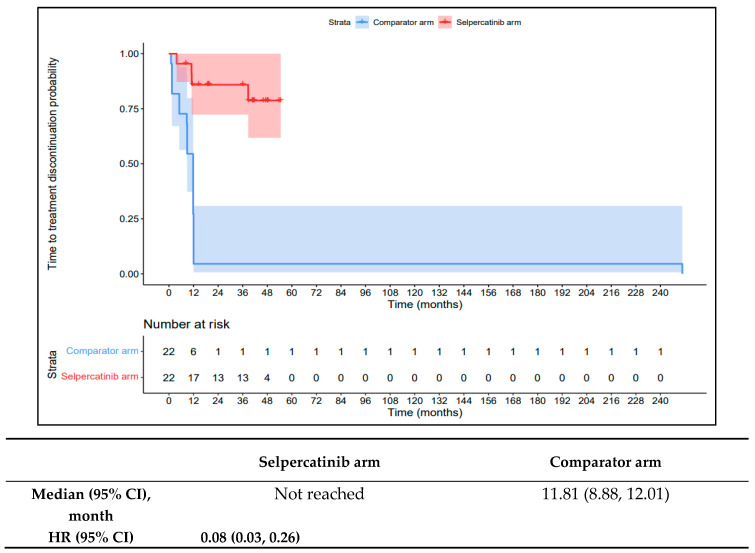
Kaplan–Meier estimates of the median TTD in the TC cohort. Abbreviations: CI, confidence interval; HR, hazard ratio; and TC, thyroid cancer. Note: The text in bold indicates statistical significance.

**Figure 10 cancers-16-00140-f010:**
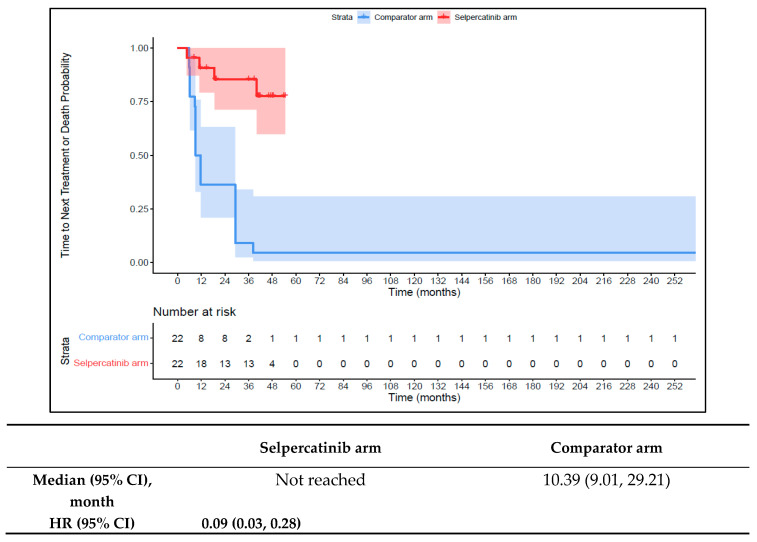
Kaplan–Meier estimates of the median TTNT-D in the TC cohort. Abbreviations: CI, confidence interval; HR, hazard ratio; and TC, thyroid cancer. Note: The text in bold indicates statistical significance.

**Figure 11 cancers-16-00140-f011:**
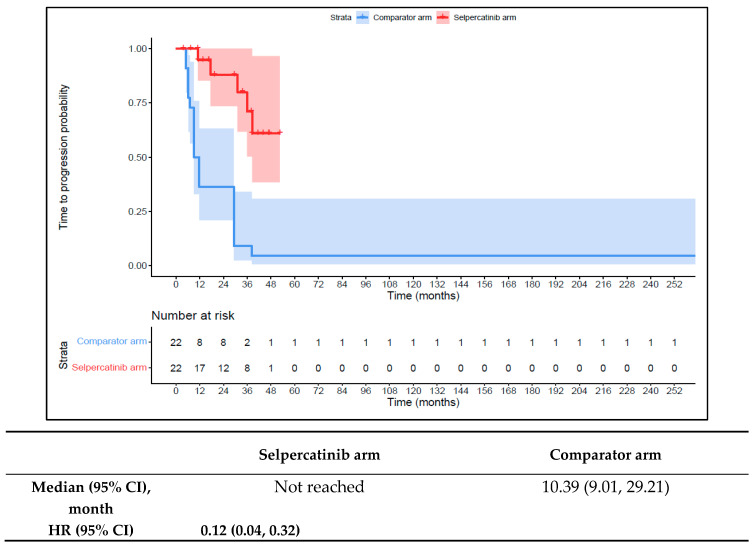
Kaplan–Meier estimates of the median TTP in the TC cohort. Abbreviations: CI, confidence interval; HR, hazard ratio; and TC, thyroid cancer. Note: The text in bold indicates statistical significance.

**Table 1 cancers-16-00140-t001:** Baseline characteristics among patients with RET fusion-positive aNSCLC.

(n {%}) Unless Specified	Pre-Matched			Genetic Matching		
	Selpercatinib Arm (n = 69)	Comparator Arm (n = 246)	*p*-Value	Selpercatinib Arm (n = 68)	Comparator Arm (n = 68)	*p*-Value
Age at index date, year						
Mean (SD)	62.1 (13.0)	56.9 (11.4)	**0.0014**	62.2 (13.1)	61.0 (10.8)	0.5836
Sex						
Female	43 (62.3)	140 (56.7)	0.4016	43 (63.2)	43 (63.2)	1.0000
Male	26 (37.7)	107 (43.3)		25 (36.8)	25 (36.8)	
Race						
White	48 (69.6)	108 (43.7)	**0.0005**	47 (69.1)	48 (70.6)	
Black/African American	4 (5.8)	12 (4.9)		4 (5.9)	4 (5.9)	0.9847
Asian	13 (18.8)	118 (47.8)		13 (19.1)	13 (19.1)	
Other	4 (5.8)	7 (2.8)		4 (5.9)	3 (4.4)	
Time from initial diagnosis to 1L start						
Q1	1 (1.5)	81 (32.8)		1 (1.5)	1 (1.5)	
Q2	6 (8.7)	70 (28.3)	**<0.0001**	6 (8.8)	6 (8.8)	0.9870
Q3	23 (33.3)	53 (21.5)		23 (33.8)	25 (36.8)	
Q4	39 (56.5)	42 (17.0)		38 (55.9)	36 (52.9)	
Stage at initial diagnosis						
Stage I-IIIA	3 (4.4)	12 (4.9)		3 (4.4)	3 (4.4)	1.000
Stage IIIB-IV	65 (94.2)	235 (95.1)	0.1641	65 (95.6)	65 (95.6)	
Unknown	1 (1.5)	0 (0.0)		0 (0.0)	0 (0.0)	
ECOG Performance Status						
0/1	65 (94.2)	240 (97.2)	0.2352	64 (94.1)	64 (94.1)	1.000
2	4 (5.8)	7 (2.8)		4 (5.9)	4 (5.9)	
Smoking Status, no	48 (69.6)	165 (66.8)	0.6650	47 (69.1)	45 (66.2)	0.7139
Brain metastases, yes	16 (23.2)	77 (31.2)	0.1981	16 (23.5)	16 (23.5)	1.0000

Abbreviations: aNSCLC, advanced non-small cell lung cancer; ECOG, European Cooperative Oncology Group; n, number of patients in sub-group; Q1, 1st quartile; Q2, 2nd quartile; Q3, 3rd quartile; Q4, 4th quartile; 1L, first-line therapy; and SD, standard deviation. Notes: Statistical comparisons between treatment arms were computed using *t*-tests for continuous variables and Chi-square and Fisher’s exact test for categorical variables. Bold text indicates statistically significant differences between the treatment arms.

**Table 2 cancers-16-00140-t002:** Baseline characteristics among patients with *RET*-mutated MTC.

(n {%}) Unless Specified	Pre-Matched			Genetic Matching		
	Selpercatinib Arm (n = 143)	Comparator Arm (n = 136)	*p*-Value	Selpercatinib Arm (n = 138)	Comparator Arm (n = 138)	*p*-Value
Age at index date, year						
Mean (SD)	55.7 (15.2)	53.8 (15.8)	0.3091	55.5 (15.4)	55.2 (13.1)	0.8554
Sex						
Female	60 (42.0)	48 (35.3)	0.2534	58 (42.0)	54 (39.1)	0.6239
Male	83 (58.0)	88 (64.7)		80 (58.0)	84 (60.9)	
Race						
White	124 (86.7)	123 (90.4)	0.2886	120 (87.0)	121 (87.7)	
Black/African American	2 (1.4)	1 (0.7)		2 (1.5)	2 (1.5)	0.9951
Asian	8 (5.6)	2 (1.5)		8 (5.8)	7 (5.1)	
Other	9 (6.3)	10 (7.4)		8 (5.8)	8 (5.8)	
Time from initial diagnosis to 1L start						
Q1	27 (18.9)	42 (30.9)		26 (18.8)	25 (18.1)	
Q2	32 (22.4)	39 (28.7)	**0.0062**	31 (22.5)	32 (23.2)	0.9961
Q3	37 (25.9)	32 (23.5)		36 (26.1)	35 (25.4)	
Q4	47 (32.9)	23 (16.9)		45 (32.6)	46 (33.3)	
Stage at initial diagnosis						
Stage I-IIIA	5 (3.5)	7 (5.2)		5 (3.6)	5 (3.6)	1.000
Stage IIIB-IV	134 (93.7)	126 (92.7)	0.7607	133 (96.4)	133 (96.4)	
Unknown	4 (2.8)	3 (2.2)				
ECOG performance status						
0/1	137 (95.8)	126 (92.7)	0.2569	132 (95.7)	131 (94.9)	0.7763
2	6 (4.2)	10 (7.4)		6 (4.4)	7 (5.1)	
Smoking Status, no	81 (56.6)	86 (63.2)	0.2053	78 (56.5)	80 (58.0)	0.8077
Brain metastases, yes	3 (2.1)	10 (7.4)	0.0374	3 (2.2)	2 (1.5)	0.6518

Abbreviations: ECOG, European Cooperative Oncology Group; MTC, medullary thyroid cancer; n, number of patients in sub-group; Q1, 1st quartile; Q2, 2nd quartile; Q3, 3rd quartile; Q4, 4th quartile; 1L, first-line therapy; and SD, standard deviation. Notes: Statistical comparisons between treatment arms were computed using *t*-tests for continuous variables and Chi-square and Fisher’s exact test for categorical variables. Bold text indicates statistically significant differences between the treatment arms.

**Table 3 cancers-16-00140-t003:** Baseline characteristics among patients with *RET* fusion-positive TC.

(n {%}) Unless Specified	Pre-Matched			Genetic Matching		
	Selpercatinib Arm (n = 24)	Comparator Arm (n = 41)	*p*-Value	Selpercatinib Arm (n = 22)	Comparator Arm (n = 22)	*p*-Value
Age at index date, year						
Mean (SD)	57.7 (16.5)	51.4 (20.2)	0.1996	56.6 (16.6)	56.1 (23.6)	0.9331
Sex						
Female	10 (41.7)	23 (56.1)	0.2614	9 (40.9)	10 (45.5)	0.7609
Male	14 (58.3)	18 (43.9)		13 (59.1)	12 (54.5)	
Race						
White	18 (75.0)	24 (58.5)	**0.0196**	18 (81.8)	20 (90.9)	
Black/African American	0 (0.0)	3 (7.3)		0 (0.0)	0 (0.0)	0.5754
Asian	1 (4.2)	12 (29.3)		1 (4.6)	1 (4.6)	
Other	5 (20.8)	2 (4.9)		3 (13.6)	1 (4.6)	
Time from initial diagnosis to 1L start						
Q1	1 (4.2)	15 (36.6)		1 (4.6)	1 (4.6)	
Q2	3 (12.5)	14 (34.2)	**0.0003**	3 (13.6)	3 (13.6)	0.9904
Q3	11 (45.8)	5 (12.2)		10 (45.5)	9 (40.9)	
Q4	9 (37.5)	7 (17.1)		8 (36.4)	9 (40.9)	
Stage at initial diagnosis						
Stage I-IIIA	0 (0.0)	3 (7.3)		0 (0.0)	0 (0.0)	
Stage IIIB-IV	24 (100.0)	36 (87.8)	0.2049	22 (100.0)	22 (100.0)	
Unknown	0 (0.0)	2 (4.9)		0 (0.0)	0 (0.0)	
ECOG Performance Status						
0/1	23 (95.8)	38 (92.7)	0.6100	21 (95.4)	19 (86.4)	0.2943
2	1 (4.2)	3 (7.3)		1 (4.6)	3 (13.6)	
Smoking Status, no	13 (54.2)	28 (68.3)	0.1873	11 (50.0)	12 (54.5)	0.7628
Brain metastases, yes	1 (4.2)	12 (29.3)	**0.0146**	1 (4.6)	0 (0.0)	0.3117

Abbreviations: ECOG, European Cooperative Oncology Group; n, number of patients in sub-group; Q1, 1st quartile; Q2, 2nd quartile; Q3, 3rd quartile; Q4, 4th quartile; 1L, first-line therapy; SD, standard deviation; and TC, thyroid cancer. Notes: Statistical comparisons between treatment arms were computed using *t*-tests for continuous variables and Chi-square and Fisher’s exact test for categorical variables. Bold text indicates statistically significant differences between the treatment arms.

## Data Availability

The data presented in this study are from the LIBRETTO-001 trial (NCT03157128) and are available on request from the corresponding author. Access will be provided after a proposal has been approved by an independent review committee identified for this purpose and after receipt of a signed data-sharing agreement.

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
