# Peer review of "Comparative Effectiveness of First-Line Selpercatinib versus Standard Therapies in Patients with RET-Activated Cancers: An Exploratory Interpatient Analysis of LIBRETTO-001"

_cancers, 2023, doi:10.3390/cancers16010140_

Round 1
Reviewer 1 Report
Comments and Suggestions for Authors
This manuscript falls under the Sensors scope and presents findings of research titled “Comparative effectiveness of first-line selpercatinib versus standard therapies in patients with RET-activated cancers: an exploratory interpatient analysis of LIBRETTO-001”. The manuscript consists of 21 pages, including 11figure, 3 tables and the list of total 29 literature references. The paper presents interesting results as well as an inquisitive and reliable interpretation of the research results. The topic original and relevant in the field of study. The Summary and Abstract provides the highlights of the key contents of the main text. The Introduction provides enough background information to justify the study. The Results are consistent with the declared methodology, presented clearly enough, supported by the tables and figure. Researchers compared investigator-assessed out comes among patients with RET-altered solid tumors who were treated with selpercatinib but naïve to standard therapies versus those previously treated with just standard therapies, and concluded that selpercatinib use as frontline therapy is associated with improved outcomes versus standard therapies in patients with RET-altered cancers in the lung, thyroid, and other non-lung, non-thyroid organs in the body. The methodology adequately described and conclusion consistent with the evidence and arguments presented. The references are appropriate and relevant to the research. However, minor typographical and grammatical errors need addressing.
Comments on the Quality of English LanguageMinor typographical and grammatical errors need addressing.
Author Response
Thank you for your review and comments. As per your suggestion, we have completed an editorial check of the manuscript draft to resolve any typographical and/or grammar errors.
Reviewer 2 Report
Comments and Suggestions for Authors
The authors performed the post-hoc comprehensive effectiveness analysis from LIBRETTO-001 study for comparing investigator-assessed outcomes among patients with RET-altered solod tumors who were treated with selpercatinib but naïve to standard therapies versus those previously treated with just standard therapies. Then, they concluded that first-line selpercatinib use is associated with improved outcomes compared to first-line standard therapies for patients with advanced/metastatic RET-activated non-small cell lung cancer (NSCLC), medullary thyroid cancer (MTC), and thyroid cancer (TC).
These are wonderful findings that warrant publication of this manuscript.
Comments:
· I agree the authors comment that there is a continued need to assess the effectiveness of selpercatinib in the first-line setting compared to the other standard therapies.
· In the section of Research ethics, the authors described that “secondary use of data from LIBRETTO-001”. Are all patient data, which included selpercatinib arm and comparator arm, from LIBRETTO-001? Moreover, they described that “patient consent was waived as this was a non-interventional study based on secondary use of data from LIBRETTO-001 and institutional review board approval was not needed.” Might it be necessary to do Opt-out procedure, isn’t it?
Author Response
Thank you for the review and comment. All data in this study (both study arms) were from the LIBRETTO-001 trial (NCT03157128). In the LIBRETTO-001 trial, patient data were collected after patients provided consent. However, since our study used previously collected data and was non-interventional in nature, no additional consent was needed. As the data were deidentified to protect privacy, a formal consent to release information was not required. There was no opt-out procedure. We have made minor revisions in the 'Informed Consent Statement' section in the manuscript.
Reviewer 3 Report
Comments and Suggestions for Authors
Selpercatinib is indicated for locally advanced/metastatic RET-activated solid tumors after progression or following prior systemic therapies. There are limited studies comparing selpercatinib to standard treatments in previously treated patients with certain solid cancers that have alterations in the RET receptor. The aim of this post-hoc comparative effectiveness analysis study was to compare investigator-assessed outcomes among patients with RET-altered solid tumors who were treated with selpercatinib but to standard therapies versus those previously treated with just standard therapies
This study showed that selpercatinib use as frontline therapy is associated with improved outcomes versus standard therapies in patients with RET-altered cancers in the lung, thyroid, and other non-lung, non-thyroid organs in the body. First-line selpercatinib use is associated with improved outcomes compared to first-line comparator therapies for patients with advanced/metastatic RET-activated cancers
Selpercatinib use was associated with a significantly lower risk of treatment discontinuation, starting a new systemic treatment (or death), or disease progression compared to standard therapies. First-line selpercatinib use is associated with improved outcomes compared to first-line comparator therapies for patients with advanced/metastatic RET-activated cancers
Overall the study is well written and ready for publication. One comment is that the testing methods used to detect RET alterations are not discussed . That is, the specific RET alterations detected are missing. The different fusion partners would be useful information that would broaden the scope of the study .
Comments on the Quality of English Language
English is fine.
Author Response
Thank you for your comment. In this manuscript, we decided to focus on selected covariates/baseline characteristics to be adjusted in order to minimize the bias due to comparing potentially prognostically different patient cohorts. From the statistical methods perspective, the number of covariates that can be included in the analysis was limited due to small sample size. While expanding discussion to testing methods and fusion partners/mutation types in the current manuscript would be of general interest, we think that focusing on describing and discussing covariates used in the statistical analysis would benefit readers the most.
Information on RET fusion partners in LIBRETTO-001 are available from other studies. For example, Efficacy of Selpercatinib in RET Fusion–Positive Non–Small-Cell Lung Cancer | NEJM, Efficacy of Selpercatinib in RET-Altered Thyroid Cancers | NEJM, Selpercatinib in Patients With RET Fusion–Positive Non–Small-Cell Lung Cancer: Updated Safety and Efficacy From the Registrational LIBRETTO-001 Phase I/II Trial | Journal of Clinical Oncology (ascopubs.org)
These studies show that a majority of patients included in LIBRETTO-001 were tested using next generation sequencing (NSG), either on tissue or blood, and the same fusion partners and mutation types were commonly observed in both treatment-naïve and pre-treated patient cohorts in similar proportions. Thank you.